Effects of gerbil disturbance on the ecological stoichiometric characteristics and nutrient uptake and utilization of H. ammodendron

Shi Jinshun 1
Pang Shuyue 1
Hao Xingming 2
Liu Hongling 3
Zhuang Li 18297081587@163.com 1
Wang Zhongke 20222006002@stu.shzu.edu.cn 1
1 College of Life Sciences, Shihezi University , Shihezi , Xinjiang , China
2 Chinese Academy of Sciences, Xinjiang Institute of Ecology and Geography , Wulumuqi , Xinjiang , China
3 Chengdu Normal University, College of Chemistry and Life Science , Chengdu , Sichuan , China
Balseiro Esteban
Electronic publication date: 2025 Apr 18
Publication date: 2025
Volume: 13
Electronic Location ID: e19287
Received 2024 Nov 13; Accepted 2025 Mar 18
Copyright: ©2025 Shi et al.
Copyright year: 2025
Copyright holder: Shi et al.
License: This is an open access article distributed under the terms of the Creative Commons Attribution License, which permits unrestricted use, distribution, reproduction and adaptation in any medium and for any purpose provided that it is properly attributed. For attribution, the original author(s), title, publication source (PeerJ) and either DOI or URL of the article must be cited.
License URL: https://creativecommons.org/licenses/by/4.0/

Keywords: Haloxylon ammodendron, Great gerbil, Stoichiometric ratio, Nutrient absorption coefficient, Photosynthetic nitrogen utilization rate, Photosynthetic phosphorus utilization rate

Funding: The Open Foundation of the State Key Laboratory of Desert and Oasis Ecology KH0054 This work was supported by the Open Foundation of the State Key Laboratory of Desert and Oasis Ecology (KH0054). The funders had no role in study design, data collection and analysis, decision to publish, or preparation of the manuscript.

==============================
Rodent activity is an important factor that affects the growth and development of Haloxylon ammodendron. Studying the effect of rodent disturbance on plant ecological stoichiometric ratios helps evaluate the mechanism by which rodent disturbance affects plant growth and development. In this study, H. ammodendron, a dominant plant, and the gerbil, a typical rodent in the Gurbantunggut Desert, were selected as research objects. By measuring the biomass, root soil , and C: N: P ecostoichiometric ratios of the assimilated branches of H. ammodendron at different growth phases, the impact of great gerbil disturbance on the biomass, ecostoichiometric ratios, and nutrient uptake and use of H. ammodendron were investigated at different growth stages. The results showed that the gerbil disturbance increased the biomass of the aboveground part of the adult H. ammodendron. Gerbil disturbance also increased the soil N/P around the roots during the growth stage and the assimilation branch when the plants were middle-aged. In addition, this disturbance decreased the C/N value. The photosynthetic nitrogen use efficiency (PNUE) and photosynthetic phosphorus use efficiency (PPUE) of H. ammodendron during various growth periods decreased, and the absorption of total nitrogen (TN) in the soil decreased. However, soil total potassium (TK) absorption increased. The soil TN absorption capacity was weakened by gerbil disturbance. Meanwhile, the TK absorption capacity was enhanced, and the biomass of adult H. ammodendron increased. PNUE and PPUE of H. ammodendron were decreased by gerbil interference. In this study, the influence of gerbil disturbance on nutrient absorption by H. ammodendron and use of H. ammodendron was determined. This has provided a baseline for further studies on the coexistence mechanisms of gerbils and H. ammodendron.

Introduction

In ecosystems, plants and herbivores interact with each other and have complex relationships. Herbivores obtain nutrients by feeding on plants. When plants are fed at a certain threshold, they maintain their growth and resist external interference through certain physiological regulations (Santamaria & Martínez, 2013; Xiao et al., 2017). Herbivore activities affect the soil nutrient content of the plant area but also change the plant’s ability to store, absorb, and use external nutrients. H. ammodendron grows mainly in desert areas, and is mainly distributed in Xinjiang, Qinghai, Gansu and other regions in China. It has high drought resistance and cold resistance. Therefore, it is essential for maintaining the ecological balance of the desert. The great gerbil (Rhombomys opimus) is a herbivorous rodent that feeds on plants such as H. ammodendron and Tamarix hispida in the desert (Yang, Liu & Huang, 2011; Zhao et al., 2021). Their biological behavior of these great gerbils have different effects on the nutrient absorption capacity of the individual and the entire forest.

Biomass is a key index reflecting the material cycle of an ecosystem (Wei, You & Zhang, 2011). During shrub biomass research, to prevent the destruction of the ecological environment, most studies have used growth indicators of shrubs to establish a mathematical model for biomass prediction to estimate shrub biomass. Ecological stoichiometry is often used to determine the responses of different plant taxa to environmental disturbances (Cheng et al., 2010; Ren et al., 2007). The soil carbon–nitrogen ratio (C/N) and carbon–phosphorus ratio (C/P) are indicators of nitrogen and phosphorus mineralization, respectively (Wang, Wang & Han, 2013; Wu, Han & Chang, 2007). The lower the soil C/N and C/P, the higher the soil mineralization. The soil N/P ratio (N/P) reflects the soil organic matter composition, external disturbance, and nutrient cycling capacity. The disturbance of herbivorous rodents changes the soil’s physical properties (Ren et al., 2021), thus affecting the soil microbial biomass and its ecological stoichiometric ratio (Pan et al., 2023). The soil microbial biomass carbon, nitrogen, and phosphorus may also be affected (Joshi & Chandra, 2023; Singh & Gupta, 2018). Herbivore interference also affects the ratio of restricted elements in plants. C/N and C/P can reflect survival strategy and growth rate (Sun et al., 2018), whereas N/P can determine the restricted elements in plants. Plant growth and nutrient use under changes in soil N and P availability (Elser & Fagan, 2010).

Photosynthetic N and P use efficiency (PNUE and PPUE, respectively) are indices used to evaluate the photosynthetic capacity of plants (Li et al., 2016; Mao et al., 2018). Gerbil interference not only changes the soil nutrient content and plant nutrients but also changes the gas exchange parameters. This can affect the ecological stoichiometric ratio, soil, and H. ammodendron pant photosynthesis (Shi et al., 2024). Meanwhile, the elemental ratio in plants often changes because of the interference of herbivores, and the plant nitrogen and phosphorus contents are positively correlated with the number of herbivores. Other nutrients may also play a role in this situation (Joern, Provin & Behmer, 2012). Moderate disturbances in animals are conducive to maintaining ecological balance, whereas excessive disturbances destroy the ecosystem (Dai et al., 2022; Zhao et al., 2021).

As the main herbivorous rodents in desert areas, great gerbils have a considerable effect on desert ecosystems. Gerbil disturbance can change soil and plant nutrients as well as photosynthetic capacity of H. ammodendron. Therefore, this study investigated the biomass, nutrient absorption coefficient, and ecological stoichiometric ratio of H. ammodendron during various phases of growth at the southern margin of the Gurbantunggut Desert, to explore the differences of the following indexes of H. ammodendron at various growth stages under the disturbance of great gerbils: (1) Differences in stoichiometric ratios of C, N, P in plants? (2) The difference of C, N, P stoichiometric ratios in the surrounding soils? (3) What are the differences in the nutrient absorption and utilization? The results of this study are helpful for better understanding desert ecosystem structure and function and provide a scientific basis for desert ecosystem management.

Materials & Methods

Study area

The study area was located at the southern edge of the Gurbantunggut Desert, Xinjiang Uygur Autonomous Region, China (44°15′N−46°50′N, 84°50′E−91°20′E). The average annual temperature in this area is 7.19 °C, the average annual wind speed is 11.17 m/s, the average annual precipitation is approximately 70–150 mm, and the average annual evaporation is more than 2,000 mm. The area has a typical continental desert climate. The interior of the desert is dominated by fixed or semi-fixed dunes. H. ammodendron is one of the dominant species in the plant community of the Gurbantungut Desert, and the great gerbils is one of the typical rodents.

Experimental design

In August 2023, after field investigation in the study area, three 50  ×  50 m rat burrow quadrats and control quadrats were set up in the areas with and without gerbil activity, respectively. Gerbil activity was determined by the absence of fresh feces or food residues and other traces of gerbil activity near the mouth of the gerbil burrow (Tchabovsky et al., 2001). The distance between the various sites was more than 500 m. Three sub-quadrats of 10  ×   10 m were arranged in each quadrat, and the distance between each sub-quadrat could not be less than 20 m. Each quadrangle contained at least 10 plants in three growth stages: young, middle-aged, and adult. According to the classification of Zhang et al. (2016) and Luo et al. (2017), Haloxylon ammodendron was divided into three growth stages by plant height and base diameter: young (plant height: 30–100 cm; base diameter: 0–2 cm); middle age (plant height: 100–200 cm; 2–4 cm); mature age (plant height: >200 cm; Base diameter 6–8 cm).

Determination of biomass

To prevent the destruction of H. ammodendron forests, the following biomass calculation model was used to estimate the biomass of H. ammodendron. In each quadrat, three plants with similar growth conditions at different phases were selected. The height H (cm), crown length d1 (cm) from east to west, and length d2 (cm) from north to south of each plant were measured using a tape measure. The measured value was the maximum length of the plant from east to west and north to south. The basal stem R (cm) and height H (cm) of each plant were measured.

The following models were used to predict the aboveground biomass (dry weight) of natural Haloxylon ammodendron (Li et al., 1995; Dang et al., 2016): (1) W=5.27CH ˆ0.794

(2) C=π×d1/2×d2/2.

The relationship between the aboveground and underground biomass dry weight of H. ammodendron is as follows:

W underground = 0.9048+0.8119W above ground

In the model, W is biomass; H is the height (cm); C is the crown width (cm2).

Determination of stoichiometric ratios of carbon (C), nitrogen (N) and phosphorus (P) in plant and soil

According to the methods of Shi et al. (2024), soil and plant samples were collected, and the carbon, nitrogen, phosphorus, and potassium contents in the assimilated branches of H. ammodendron were determined using the Bao (2008) nutrient content determination method. Soil carbon, nitrogen, and phosphorus contents were determined using the same method. Plant C/N, C/P, N/P, and soil C/N, C/P, and N/P ratios were obtained from the measured data.

Determination of PPUE and PNUE

According to Shi et al.’s (2024) method, the net photosynthetic rate of H. ammodendron in the borrowed and control areas was determined. Six plants with relatively uniform growth were selected for each growth stage and healthy assimilation branches were selected for determination. Three values were obtained for each measurement and each parameter was based on an average value. Photosynthetic nitrogen use efficiency (PNUE) and photosynthetic nitrogen use efficiency (PNUE) were calculated based on the obtained data and the following formula: (3) PNUE=Pn/Nmass

(4) PPUE=Pn/Pmass

where Pn represents the net photosynthetic rate and Nmass represents the assimilated branch nitrogen content per unit mass. Pmass represents the content of phosphorus per unit mass.

Determination of biological absorption coefficient

The biological absorption coefficient (A) reflects the ability of plants to absorb and accumulate chemical elements from the environment. To effectively quantify the absorption capacity of plants for each nutrient in the soil, we calculated the absorption coefficients for carbon, nitrogen, phosphorus, potassium, Asoc, ATN, ATP, and ATK. The formula is as follows: (5) A=Lx/Nx

where Lx is the content of x element in each part of the plant body, and Nx is the content of x element in the corresponding soil (Liu et al., 2010).

Statistical analysis

Microsoft Excel was used to calculate and organize the data. Two-factor ANOVA was used to analyze the effects of growth stage, presence or absence of gerbils, and their interactions on the ecological stoichiometric ratio of soil and plants around the roots, nutrient absorption coefficient, and photosynthetic nitrogen and phosphorus use. An independent samples t-test was used to compare all indices measured in this study with or without gerbil disturbance. Post hoc Duncan’s multiple comparison test was used to examine the differences in biomass, nutrient absorption coefficient, net photosynthetic rate, and photosynthetic nitrogen and phosphorus use at different growth stages. Pearson’s correlation analysis was used to analyze the relationship between soil nutrients and nutrient absorption coefficients as well as between photosynthetic nitrogen and phosphorus use. The significance level was P < 0.05. SPSS 23 software (SPSS Inc., Chicago, IL, USA) was used for data analysis. Graphics were drawn with OriginPro 2024 (OriginLab Corp., Northampton, MA, USA).

Results

Effects of gerbil disturbance on the biomass of H. ammodendron

The growth index and biomass of H. ammodendron at various phases of growth were affected by biological activity of the great gerbils. As shown in Table 1, when there was no significant difference in plant height and base diameter, gerbil disturbance had no significant effect on the crown width of young and middle-aged adults, but significantly increased the crown width of adults (P < 0.05). Gerbil disturbance had no significant effect on the aboveground and underground biomass of young and middle-aged adults but significantly increased the aboveground and underground biomass of adults (P < 0.05) (Fig. 1). Gerbil disturbance significantly increased the crown width and aboveground and underground biomasses of adult H. ammodendron (P < 0.05).

Table 1 Growth indexes of H. ammodendron at different growth stages.

Growth stage	Presence or absence of gerbils	High (cm)	Basal diameter (cm)	Crown breadth (cm2)	
Youth	Control area	80 ± 14.1a	1 ± 0.5a	1,459 ± 246.7a	
Borrowed area	84 ± 10.8a	2 ± 0.5a	1,414 ± 206.7a	
Middle	Control area	174 ± 10.8a	3 + 0.0a	4,768 ± 388.9a	
Borrowed area	170 ± 11.9a	3 ± 0.5a	5,986 ± 786.7a	
Adult	Control area	325 ± 18.7a	7 ± 0.8a	13,706 ± 393.9b	
Borrowed area	320 ± 12.2a	7 ± 0.8a	18,682 ± 2635.5a	
Notes.

Different lowercase letters indicate that there is a significant effect of the gerbil disturbance on H. ammodendron.

P < 0.5 indicates a significant difference.

Figure 1 Effect of disturbance of great gerbils on the biomass of H. ammodendron.

Different uppercase letters indicate that there are significant differences in the indexes of H. ammodendron at each growth stage (P < 0.5), and different lowercase letters indicate that there is a significant influence on H. ammodendron with or without great gerbil disturbance. P < 0.5 indicates a significant difference.

Effects of gerbils disturbance and growth stage and their interaction on C: N: P in plants and soil around roots of H. ammodendron

As shown in Table 2, the presence or absence of gerbil disturbances had significant effects on plant N/P, soil C/N, and soil N/P (P < 0.05). Except for plant C/P, plant C/N, plant N/P, and peri-root soil ecological stoichiometric ratios of H. ammodendron at various phases of growth were significantly different (P < 0.05). The soil C/P and N/P of H. ammodendron at various phases of growth had significant effects (P < 0.05) but had no significant effects on the other indices (P < 0.05).

Table 2 Effects of disturbance and growth stage of gerbils and their interaction on nutrient uptake and photosynthetic nitrogen and phosphorus utilization of H. ammondendron.

Index		Plant C/N	Plant C/P	Plant N/P	Soil C/N	Soil C/P	Soil N/P	
Presence or absence of gerbils	F	3.317	4.618	10.580	119.597	1.556	222.134	
P	0.094	0.053	0.007	<0.0001	0.236	<0.0001	
Growth stage	F	11.523	1.807	3.965	54.112	4.056	218.033	
P	0.002	0.206	0.048	<0.0001	0.045	<0.0001	
Interactions	F	1.599	0.808	2.068	2.803	9.681	23.275	
P	0.242	0.468	0.169	0.100	0.003	<0.0001	
Notes.

P < 0.05 was significant.

Effects of gerbils disturbance on the stoichiometric ratio of soil and assimilating branches around the roots of H. ammodendron

As shown in Fig. 2, the soil C/N ratio of H. ammodendron at all stages and the soil C/P ratio of H. ammodendron at all adult stages were significantly reduced by the disturbance by great gerbils. The soil N/P ratio of H. ammodendron at all stages was significantly increased (P < 0.05). Soil N/P values were 29.16, 14.82, and 51.50 in young, middle, and adult rats, respectively. Meanwhile, those in the control were 9.98, 8.55, and 27.16, respectively. The plant C/N of the assimilated branches was significantly decreased and the plant N/P was significantly increased (P < 0.5). However, there was no significant effect on other ecological stoichiometric ratios. In the borrowed area, the plant N/P ratios of the young, medium, and adult H. ammodendron were 21.51, 26.92, and 24.97, respectively. In the control area, the N/P ratios of young, middle-aged, and adult plants were 18.72, 19.51, and 23.28, respectively. Plant and soil N/P ratios were improved by gerbil disturbances.

Figure 2 Effects of great gerbil disturbance on nutrient stoichiometric ratios in soil and assimilated branches of H. ammodendron.

(A–F) The influence of gerbil disturbance on the stoichiometric ratio of nutrients in soil and assimilated branches of H. ammondendron. An asterisk (*) indicates significant difference (P < 0.05).

Effects of gerbil disturbance and growth stage on nutrient absorption coefficient and photosynthetic nitrogen and phosphorus use efficiency of H. ammodendron

As shown in Table 3, gerbil disturbance and growth stage significantly affected the absorption of TN and TK by H. ammodendron. Under the influence of gerbil disturbance and growth stages, the absorption of TN, TP, and TK by H. ammodendron was significantly different (P < 0.05). Meanwhile, gerbil disturbance and growth stages had significant effects on PNUE (P < 0.05). The gerbil disturbance and the combined effects of growth stages had significant effects on PNUE, but no significant effects on PPUE.

Table 3 Effects of disturbance and growth stage of gerbils and their interactions on nutrient uptake and photosynthetic nitrogen and phosphorus utilization of H. ammodendron.

Index		ASOC	ATN	ATP	ATK	PNUE	PPUE	
Presence or absence of gerbils	F	2.587	46.561	0.022	1,361.305	794.898	0.079	
P	0.134	<0.0001	0.884	<0.0001	<0.0001	0.783	
Growth stage	F	1.157	58.169	1.157	13.398	119.454	1.424	
P	0.347	<0.0001	0.347	0.001	<0.0001	0.279	
Interactions	F	3.848	4.655	0.013	94.365	54.343	0.631	
P	0.051	0.032	<0.0001	<0.0001	<0.0001	0.549	
Notes.

P < 0.05 was significant.

Effects of gerbils disturbance on nutrient uptake and photosynthetic nitrogen and phosphorus use efficiency of H. ammodendron

By influencing the soil nutrients around the roots and nutrients in the assimilating branches, gerbils changed the nutrient absorption coefficient of H. ammodendron and had different effects on the nutrient absorption and use of H. ammodendron at various phases of growth. As shown in Fig. 3, gerbil disturbance significantly reduced the absorption of TN from the soil by H. ammodendron at each stage but significantly increased the absorption of TK by H. ammodendron at each stage (P < 0.05). For middle-aged H. ammodendron, the absorption capacity of TP by assimilating the branches of H. ammodendron significantly increased (P < 0.05). However, there was no significant effect on SOC absorption of H. ammodendron at any stage. Meanwhile, the photosynthetic nitrogen use efficiency (PNUE) and photosynthetic phosphorus use efficiency (PPUE) of H. ammodendron at each stages were significantly decreased by the disturbance of gerbils (P < 0.05) and had significant effects on young, middle, and adult H. ammodendron (P < 0.05).

Figure 3 (A–F) Changes of biological absorption coefficient and photosynthetic nitrogen and phosphorus utilization rate of H. ammondendron with or without gerbil disturbance.

Different uppercase letters indicate that there are significant differences in the indexes of H. ammondendron at each growth stage (P < 0.5), and different lowercase letters indicate that there is a significant influence on H. ammondendron with or without great gerbil disturbance (P < 0.5).

Correlation between soil nutrients and their stoichiometric ratios and nutrient uptake and utilization of H. ammodendron

As shown in Fig. 4, gerbil disturbance significantly changed the correlation between the soil and its stoichiometric ratio around the roots of H. ammodendron, its biological absorption coefficient, and photosynthetic nitrogen and phosphorus use. In the control group, ASOC was significantly negatively correlated with all indices except soil C/N, ATN was significantly negatively correlated with all indices except soil C/N and soil C/P, ATP was significantly negatively correlated with TN, TP, and soil N/P. PNUE was significantly negatively correlated with SOC and soil C/P. Soil C/N was positively correlated with the ASOC, ATN, and ATP levels (P < 0.05). In the rat-hole area, ASOC was significantly negatively correlated with SOC and soil C/P, ATN was significantly negatively correlated with TN, and soil N/P and was significantly positively correlated with soil C/N, ATP was significantly negatively correlated with TP, and ATK and PNUE were significantly positively correlated with soil C/P (P < 0.05). Therefore, gerbil disturbances can alter the absorption and use of soil nutrients by H. ammodendron.

Figure 4 Pearson correlation analysis of soil nutrients and their stoichiometric ratios with nutrient uptake and photosynthetic nitrogen and phosphorus utilization of H. ammodendron.

(A) Control area (no gerbil disturbance); (B) rat hole area (with gerbil disturbance), an asterisk (*) indicates a significant correlation (P < 0.05).

Discussion

Effects of the disturbance of great gerbils on the biomass of H. ammodendron

Biomass refers to the amount of individual, population, or community material. Studying the biomass of H. ammodendron can be used to predict and evaluate its ecological function and desert productivity (Li et al., 2018). In this study, according to the biomass estimation model established by Dang et al. (2016), the biomass of H. ammodendron at each growth stage under the disturbance of great gerbils was estimated. Gerbil disturbance significantly increased the crown width and biomass of the aboveground and subsurface portions of the adult H. ammodendron (P < 0.05), which was similar to the study by Xiang, Wang & Lv (2020). As one of the main rodents at the southern margin of the Gurbantunggut Desert, the activities of great gerbils are not limited to feeding and nesting. Their burying behavior may improve soil permeability, promote root growth and water absorption, and indirectly increase the biomass of adult H. ammodendron. Changes in biomass may affect niche competition and interspecific relationships in H. ammodendron. An increase in the biomass of mature H. ammodendron may give it a dominant position in resource competition, thus affecting the growth and distribution of other plant populations. We found that the gerbil mainly lived in the roots of adult H. ammodendron. The growth indicators and biomass of young and middle-aged H. ammodendron were not significant, but the crown width and biomass of adult H. ammodendron were significant (P < 0.05).

Effects of gerbils disturbance, growth stage and interaction on C: N: P of H. ammodendron

The soil nutrient stoichiometric ratio can be used to evaluate the soil nutrient supply capacity and quality (Zhao & Huang, 2022). C/N is related to the decomposition rate of soil organic matter (Wang & Yu, 2008). Meanwhile, C/P reflects the mineralization capacity of phosphorus in the soil. A lower C/P ratio can promote the decomposition of organic matter and release other nutrients in the soil (Zeng et al., 2015; Tao et al., 2017). In this study, soil C/N and N/P were significantly affected by gerbil disturbances. Meanwhile, soil C/N, C/P, and N/P at different growth stages were significantly different. There were significant differences in the soil C/P and N/P ratios between the two groups (P < 0.05). The soil elements around the roots of H. ammodendron at various phases of growth change correspondingly under the disturbance of gerbils. This may be related to most gerbils and roots of adult H. ammodendron. Therefore, the data were further analyzed using a t-test and one-way variance comprehensive analysis. Gerbil disturbance significantly reduced soil C/N at each stage, indicating that gerbil disturbance increased the decomposition rate of organic matter around the roots of H. ammodendron at each growth stage and promoted the release of nutrients. Especially for gerbils burrowing around adult H. ammodendron plants, disturbance also significantly reduced the soil C/P (P < 0.05). Therefore, gerbil disturbance was more conducive to promoting the release of soil nutrients around the roots of adult H. ammodendron. The N/P ratio reflects soil nutrient cycling capacity (Zhao, Kang & Han, 2015). Gerbil disturbance significantly increased the soil N/P at each growth stage (P < 0.05), indicating that gerbil disturbance significantly enhanced the soil nutrient cycling ability around the roots of H. ammodendron.

There is a close relationship between soil and aboveground ecological stoichiometric characteristics of plants, and the two interact and restrict each other (Jonas et al., 2010; Elser et al., 2000). The two-factor analysis of variance showed that gerbil disturbance had a significant effect on plant N/P (P < 0.05). There were significant differences in plant C/N and plant N/P at different growth stages (P < 0.05). However, the combined effect of the two had no significant effect on the ecological stoichiometric ratio of plants. This may be because, under the condition of long-term coexistence of H. ammodendron and gerbil, H. ammodendron evolved and passed on the genes adapted to gerbil interference to the next generation through long-term adaptation. Therefore, there was no significant difference in the ecological stoichiometric ratio of the plant when the gerbil interference and growth stage acted together. With the later plant growth, the gerbil directly carried out mechanical damage activities such as plant chewing. The ecological stoichiometric ratios of H. ammodendron with or without gerbil interference at different growth stages gradually showed significant differences. Plant leaf N/P ratios can be used to represent the restriction patterns of N and P in plants. When the N/P ratio was greater than 16, plant growth was prone to P restriction (Huang, Gao & Huang, 2021). Plant N/P (29.16, 14.82 and 51.50) at each growth stage in the rat hole area was significantly higher than control area (9.98, 8.55 and 27.16) and was greater than 16 (P < 0.05) (Fig. 2), indicating that the growth of H. ammodendron was more easily restricted by P owing to the disturbance of gerbils.

Effects of disturbance and growth stage and their interaction on biological absorption coefficient and photosynthetic nitrogen and phosphorus use efficiency of H. ammodendron

The biological absorption coefficient reflects the ability of a plant to absorb and accumulate chemical elements from the environment (Liu et al., 2010). In the present study, gerbil disturbances and the growth stage significantly affected the absorption of TN and TK by H. ammodendron. Under the influence of gerbil disturbances and different growth stages, the absorption of TN, TP, and TK by H. ammodendron differed significantly (P < 0.05). This may be due to the different feeding conditions of H. ammodendron at each phases. Therefore, the survival strategies of H. ammodendron at each phases are different. Gerbil disturbance significantly decreased the absorption of TN from the soil by H. ammodendron at each growth stage (P < 0.05). Xu et al. (2012) showed that the settlement of great gerbils can increase the N content in soil, which reduces the selectivity of ion absorption by H. ammodendron under high N concentrations, thus reducing the soil N absorption. Correlation analysis also showed that the absorption coefficient of soil N by H. ammodendron was inversely proportional to the content of soil N under gerbil disturbance. Gerbil disturbance significantly increased soil TK absorption by H. ammodendron at each growth stage (P < 0.05). Soil ventilation can affect root respiration and ATP supply, thereby affecting plant nutrient absorption (DeVries, Thébault & Liiri, 2013). The burrowing behavior of gerbils improved soil aeration, enhanced root respiration, and increased the ATP supply, thus increasing the absorption efficiency of K ions by plants. Compared with other ions, the absorption rate of K ions by roots is relatively high. Therefore, the activities of the gerbil change the absorption effect of H. ammodendron on ions and promote the absorption of K ions.

PNUE and PPUE reflect the relationship between plant leaf nutrient content and physiology (Amane & Kanehiro, 2011). The two-factor analysis of variance showed that the presence or absence of gerbil disturbances and different growth stages significantly affected the PNUE of H. ammodendron (P < 0.05). Gerbil disturbance and the combined effects of the growth stage had significant effects on the PNUE of H. ammodendron but had no significant effects on PPUE. This may be because the disturbance of gerbils significantly changed plant N but had no significant effect on plant P (Shi et al., 2024). The plants obtained more N to restore photosynthetic capacity by increasing PNUE after gnawing (Fernando et al., 2008). However, the PNUE of H. ammodendron was significantly reduced after the disturbance of gerbils in this study. This may be due to the long-term gnaw interference of gerbils, which made H. ammodendron synthesize defense substances rather than restoring the photosynthetic rate. The PPUE of H. ammodendron was significantly decreased by the gerbil disturbance. This may be related to the decreased PPUE caused by mechanical damage caused by the activity of the gerbil.

Conclusions

Gerbil disturbance can significantly increase the biomass of aboveground and subsurface parts of adult H. ammodendron. It can improve the decomposition rate of soil organic matter around the roots of each growth stage and enhance soil nutrient cycling. However, the growth of H. ammodendron is more susceptible to P restriction. From the perspective of nutrient absorption and use, gerbil disturbance decreased the absorption capacity of H. ammodendron for soil N and increased the absorption capacity of H. ammodendron for soil K. Gerbil disturbance significantly reduced photosynthetic nitrogen and phosphorus use in H. ammodendron. Therefore, this study concluded that, under gerbil disturbance, H. ammodendron nutrients mainly came from the absorption of soil nutrients. The findings have shown that the growth of H. ammodendron was easily restricted by photosynthesis owing to gerbil disturbance. These findings deepen our understanding of the mechanisms of plant-animal interactions in desert ecosystems, provide a scientific basis for the protection and management of desert ecosystems, and can be used to develop targeted conservation strategies for desert plant communities.

Supplemental Information

Supplemental Information 1 Raw data

Additional Information and Declarations

Competing Interests

Author Contributions

Data Availability

The authors declare there are no competing interests.

Jinshun Shi conceived and designed the experiments, performed the experiments, analyzed the data, prepared figures and/or tables, authored or reviewed drafts of the article, and approved the final draft.

Shuyue Pang conceived and designed the experiments, performed the experiments, analyzed the data, prepared figures and/or tables, authored or reviewed drafts of the article, and approved the final draft.

Xingming Hao analyzed the data, prepared figures and/or tables, authored or reviewed drafts of the article, and approved the final draft.

Hongling Liu performed the experiments, prepared figures and/or tables, authored or reviewed drafts of the article, and approved the final draft.

Li Zhuang conceived and designed the experiments, performed the experiments, analyzed the data, prepared figures and/or tables, authored or reviewed drafts of the article, and approved the final draft.

The following information was supplied regarding data availability:

The raw measurements are available in the Supplementary File.

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
