# Peer review of "Effects of gerbil disturbance on the ecological stoichiometric characteristics and nutrient uptake and utilization of H. ammodendron"

_PeerJ, doi:10.7717/peerj.19287_

## Round 0.1 · original submission · Major Revisions

I have now received the reviewers' comments and made my own analysis of the manuscript. All reviewers found that the results regarding the stoichiometric effects of the gerbils' burrowing are promising. However, essential aspects of the manuscript must be corrected before any decision can be taken (see this letter, the comments made by the reviewers and the annotated pdf).

1. The manuscript includes information already published in a previous paper. You can use this information, but you cannot include it in the methods and Results sections, you can use it as background in the Introduction and discuss the relationship between the stoichiometric implications and growth and photosynthesis, but you cannot present these data as originals as you have done (including it in Methods, Results, statistical analyses and figures).

2. The experimental design is a typical factorial design. You cannot analyze the results with t-test, this is wrong because you are misusing the errors. You need to apply a 2way-ANOVA, where the errors are considered in the whole set of comparisons. In addition, you will be able to analyze interactions among factors (gerbils X age), that seem important in your results.

So now I am giving you a particular opportunity to correct this manuscript, but you must fulfil the two aspects pointed out above, and rewrite the whole manuscript according to the new structure. In addition, check for your English that needs a deep language correction. Also please check the citations, there are many very important contributions to plant stoichiometry and animal-plant interactions.

If you decide to correct the manuscript, please include a complete list of changes and answers to each comment from each reviewer and editor.

·

Basic reporting

1. Basic Reporting
The report is well written, but many grammatical errors need linguistic review.

Experimental design

2. Experimental design
Materials & Methods
- Study area
It is preferable to put coordinates in place to define the study area better.
- Experimental design and analysis process
Clear

Validity of the findings

4. Validity of the findings
clear and correct
5. Tables and figures
clear and effective

Additional comments

6. General comments
The whole paper needs an English review.
7. Confidential notes to the editor ammodendron.
None

·

Basic reporting

Overall, the manuscript is good and well suited for publication in PeerJ.

Experimental design

Experimental design is good and can be replicated.

Validity of the findings

The findings are new and informative for readers.

Additional comments

Comments to the Author
Ecological stoichiometric ratio is an important index to reveal the response of plant species to environmental disturbance. As described in the manuscript, the study of the effect of rodent disturbance on the stoichiometric ratio of plant ecology is helpful to further evaluate the effect of rodent disturbance on the mechanism construction of plant growth and development. In this manuscript, H. ammodendron, a dominant plant and a typical rodent in Gurbantungut desert, was selected as the research object. The biomass, net photosynthetic rate, and nutrient contents in soil around roots and assimilated branches of H. ammodendron were determined to investigate the effects of the disturbance of H. ammodendron on the biomass, ecological stoichiometric ratio and nutrient uptake and utilization of H ammodendron at different growth stages. It reveals that after the interference of the gerbils, the absorption of TN by H. ammodendron in each long stage decreased but the absorption of TK increased, and the influence of great gerbils on H. ammodendron decreased the photosynthetic nitrogen and photosynthetic phosphorus utilization of H. ammodendron. This manuscript reveals the effect of the disturbance of the great gerbils on the nutrient absorption and utilization of H. ammodendron, and lays a foundation for the future study of the coexistence mechanism of the great gerbils and H. ammodendron. The manuscript is rich in content and reveals the influence of gerbil disturbance on nutrient absorption and utilization of H. ammodendron, which has novelty and practical significance. This manuscript can be accepted for publication. Some suggestions need to be revised as follows:
In line 16: The H. ammodendron abbreviation needs to complete the full name
In line 113-114: How to divide the different growth stages of H. ammodendron, Suggest further describe it in further detail.
In line 136-138: The description of sample collection and processing is not detailed and will be rewritten in further detail.
In line 64-66: Propose to reduce “Soil ecological stoichiometric characteristics …… between soil carbon (C), nitrogen (N) and phosphorus (P).”
In line 269-275: Propose to rewrite “As one of the main rodents in the southern margin……thus affecting the growth and distribution of other plant populations.” to make it more fluent.
In line 351-352: Propose to reduce “P is an important nutrient element that limits the primary productivity of vegetation in terrestrial ecosystems.” and merge with the previous paragraph.

Reviewer 3 ·

Basic reporting

1. General Comments
Overall, the manuscript presents an interesting and valuable study on the impact of gerbil interference on Haloxylon ammodendron. However, there are several aspects that need to be addressed to enhance the quality of the paper.
2. Specific Comments
2.1 Content in Introduction
The authors mention in the introduction (line 52) that gerbils also feed on the phloem of Haloxylon ammodendron, but this aspect is not reflected in the overall experimental design. It is recommended to delete the term "phloem" to maintain consistency between the introduction and the experimental part.
2.2 Data Citation in Discussion
When discussing the N/P values of the assimilating branches of Haloxylon ammodendron, the authors are advised to cite specific data from Figure 2. Incorporating such detailed data will make the discussion more persuasive and provide stronger evidence for the conclusions drawn.
2.3 Significance in Discussion
At the end of the discussion section, it is suggested to add one or two sentences to clarify the importance of the research results for the management and understanding of desert ecosystems. Elaborating on the specific application values and practical implications will enhance the broader significance of the study. For example, the results could be used to develop more targeted conservation strategies for desert plant communities or to optimize land - use planning in desert areas.
2.4 Language and Logical Flow
The transitions between paragraphs are not very smooth in terms of language. When presenting different research contents, the authors should add more connecting sentences to improve the overall logical coherence. This will help readers follow the thought process more easily and understand how different parts of the study are related.
2.5 Manuscript Details and Formatting
There are some details in the manuscript that require attention. The authors need to conduct a full - text format revision to ensure there are no formatting errors.
In line 284, a Chinese punctuation mark is used, which should be corrected to the appropriate English punctuation.
In line 332, the genus name "H" in "H. ammodendron" should be italicized to follow the standard taxonomic naming convention.
In line 332, there is a citation format issue with "(T F V D et al., 2013)". The correct citation format should be in accordance with the journal's requirements, which usually involves proper author name presentation (e.g., if there are multiple authors, the correct abbreviation and order should be followed) and a consistent style for the year and other elements.
I believe that with these revisions, the manuscript will be more suitable for publication in Peer J. The authors are encouraged to carefully consider these suggestions and make the necessary improvements.

Experimental design

no comment

Validity of the findings

no comment

Reviewer 4 ·

Basic reporting

Shi et al. investigated the disturbance impact of the rodent species great gerbil on the growth and nutritional status of a shrub species Haloxylon ammodendron by analyzing the stoichiometry of the plant species and its soil. I carefully read the whole manuscript and feel very disappointed to see the bad attitude of the authors towards this manuscript. They should restructure the whole story and put much more efforts on the writing.
1. The english language was poorly used to describe the study design and results. I strongly suggest the authors to rewrite the whole manuscript.
2. it is very hard to follow what the authors trying to introduce in terms of the background. I suggest them to conclude proper scentific questions and write a story aligned to hypotheses.
3. Redesign figures and add detailed figure legends. For example, I see the authors designed a two-factorial design (rodent disturbance and plant growth stage), but why it was not written in the text, which was not described either in the statistics.
4. Please refer to other published studies about how to write result section of a study.

Experimental design

It is difficult to comment on the experimental design as it was very shortly described in the text. In fact, the whole material and method section was not decribed enough in detail.

Validity of the findings

I don't trust the findings as the readers could not understand how they analyze the data. I suggest they used a two-way ANOVA to reanalyze the data and describe how the plant performance change ontogenetically and how such changes differ between borrowed and control plots.

Additional comments

Please show a better attitude in preparing manuscript.

---

## Round 0.2 · accepted · Accept

The authors have correctly addressed all the comments by the reviewers and the Associated Editor